# Differences in the relationships between interoceptive sensibility and self-objectification in women with high and low body dissatisfaction: A network analysis

Akansha Mahesh Naraindas[1,2]*, Amy McInerney[3], Sonya Deschênes[1], Sarah Maeve Cooney[1]

1 School of Psychology, University College Dublin, Dublin, Ireland, 2 Centre for Public Health, Queen's University Belfast, Belfast, United Kingdom, 3 Department of Population-Based Medicine, University of Tübingen, Tübingen, Germany

* a.naraindas@qub.ac.uk/akansha.maheshnaraindas@ucdconnect.ie

## Abstract

Body dissatisfaction is a normative experience with the potential to impact women's mental and physical health. It is closely related to self-objectification, where the body is viewed as an object for aesthetic evaluation rather than for its functional attributes. Self-objectification not only affects women's body image but is associated with diminished subjective awareness of, and attention towards, internal bodily states (i.e., interoceptive sensibility). This study uses network analysis and network comparison to investigate the relationships between features of interoceptive sensibility and self-objectification in women with high (N = 348) and low (N = 354) body dissatisfaction. The results firstly revealed significant differences in the connections between interoceptive sensibility and self-objectification in women with high and low body dissatisfaction (p = 0.026). High body dissatisfaction in women was characterised by lower levels of body listening, self-regulation, and body trusting. Alternatively, high emotional awareness was most central to women with low body dissatisfaction. This study highlights the central role of interoceptive sensibility in body dissatisfaction and identifies key features of interoceptive sensibility to target for improving body dissatisfaction.

## Introduction

Body dissatisfaction refers to negative perceptions and attitudes about one's external appearance [1], and is typically exemplified by shape and weight concerns [2]. While body dissatisfaction affects individuals of all ages, genders, and ethnicities, it is highly prevalent among women and is thus considered a normative discontent [3]. Affecting up to 72% of women [4], body dissatisfaction constitutes a burden at a global level, with higher levels being linked to a host of social, health (physical and

**Data availability statement:** The data presented in this study are openly available via OSF: https://osf.io/25cjh

**Funding:** This project was funded by the University College Dublin Ad Astra research fellowship (ID: C776AAB4) to S.M.C. and associated Ad Astra PhD scholarship to A.M.N. The funders had no role in study design, data collection and analysis, decision to publish, or preparation of the manuscript.

**Competing interests:** The authors have declared that no competing interests exist.

mental), and economic consequences [5,6]. A recent study estimates the financial impact of body dissatisfaction between $226.4 billion and $506.5 billion in the United States alone [7]. Feminist theories suggest that body dissatisfaction and its negative outcomes stem from women's routine encounters with objectification, which pushes them to perceive their bodies primarily as objects for external evaluation, rather than appreciating their functional capacities or internal sensations [8,9]. The ability to sense, interpret, and integrate sensations originating from within the body is known as interoception, and is vital to the maintenance of mental and physical health [10]. While current knowledge on the role of self-objectification in contributing to high levels of body dissatisfaction is considerable [11], there is a notable lack of studies investigating its relationship with interoception among women with varying degrees of body dissatisfaction, and its broader implications for health and well-being.

Interoception is the sense that allows the body to detect and respond to internal signals like hunger, thirst, body temperature, sleepiness, and emotional states such as joy, anxiety, or depression [12]. Atypical interoceptive processing contributes to increased vulnerability to psychological disorders [10], particularly in women, due to significant internal physical and hormonal changes they experience throughout development (See: Murphy et al. 2019 [13] for review). Dynamic changes in women's reproductive physiology, such as puberty, menstruation, and pregnancy, create a variable body environment, decreasing the reliability of internal signals [13–15]. Self-objectification has also been found to influence the perception of inner body signals. Indeed, women with high body dissatisfaction consistently report increased self-objectification [11] and exhibit reduced awareness of internal bodily markers of well-being such as illness, fatigue, hunger, and coldness [16–19]. One explanation, the competition of cues hypothesis [20], suggests that because attention to the body is limited, that women who focus on their external appearance have less attention available for processing internal bodily signals. For example, Ainsley and Tsakiris (2013) conducted a heartbeat detection test where participants counted their perceived heartbeats during timed intervals, and these counts were compared to their actual heartbeats. They found that women with higher self-objectification had lower sensitivity to internal body signals and were less accurate at perceiving their heartbeats [21]. Therefore, self-objectification in women with high levels of body dissatisfaction might divert attention from important interoceptive signals, potentially contributing to the health and well-being consequences of high body dissatisfaction.

McKinley and Hyde (1996) describe the process of self-objectification through the 'body conscious variables': body surveillance and body shame [8]. Body surveillance involves monitoring the body's appearance based on anticipated evaluations by others [8,22]. This causes body shame by prompting women to measure their bodies against unattainable internalised beauty ideals [8]. Self-objectification theory suggests that women with elevated levels of self-surveillance not only emphasize their body's external appearance but also prioritize it over its functionality or overall health [23]. Indeed, body surveillance has been linked to restrained eating [24,25], increased cigarette smoking [26], excessive indoor tanning [27], and willingness to undergo elective cosmetic surgery [28]. Furthermore, body surveillance is strongly

associated with body dissatisfaction [11,29] and alterations to how attentional resources are allocated towards bodily processes [21,30,31]. For example, a study by Van De Veer et al. (2015) showed that induced self-objectification from brief mirror exposure or viewing model advertisements impaired participants' real-time awareness of internal signals like hunger and satiety [32]. This aligns with the competition of cues hypothesis, highlighting an antagonism between interoceptive and exteroceptive signals.

One dimension of interoception that is a key area of focus in body image research is interoceptive sensibility: the self-reported awareness of one's internal sensations [10]. Interoceptive sensibility is commonly measured using questionnaires such as the Multidimensional Assessment of Interoceptive Awareness (MAIA-2), categorising interoceptive sensibility into several distinct subscales [33]. Recently, research has demonstrated a strong negative relationship between total MAIA-2 scores and self-objectification in women across adulthood [34] and in men [35], highlighting the consistency of this relationship across diverse demographic groups. However, due to the multidimensional nature of both interoceptive sensibility and self-objectification, approaches that employ a total score do not fully elucidate the interrelationships between their features in influencing body dissatisfaction. Interestingly, investigation into the MAIA-2 subscales has revealed relationships with both negative and positive body image [36]. Todd et al. (2019) found that the Noticing, Emotional Awareness and Trusting scales significantly predicted different facets of body dissatisfaction (appearance orientation and overweight preoccupation) [36]. The study also found that the Trusting, Emotional Awareness and Noticing scales predicted features of positive body image (body appreciation and body functionality). This indicates that different dimensions of interoceptive sensibility are related to both high and low body dissatisfaction.

Recently, research has begun to view the features of self-objectification theory and interoceptive sensibility as part of the same network of interacting influence [37]. Network analysis allows us to model these individual features (e.g., the MAIA-2 individual sub-scales and self-objectification (body shame and surveillance)) as an interactive network, examining which individual features exert the most influence on the network and which are most strongly connected to other features of a different construct. Networks consists of *nodes*, which symbolize variables (e.g., individual features of interoceptive sensibility and self-objectification), and *edges*, representing the links between them [38]. Through network analysis, it is possible to identify which components exert the greatest influence within the network (i.e., centrality) and to visualise relationships among these nodes [39].

Additionally, whilst body dissatisfaction ranges from non-existent to severe [40], much of the existing research has focused on the severe end, particularly high body dissatisfaction. Examining and comparing the relationships between interoceptive sensibility and self-objectification in women with both high and low body dissatisfaction, could be useful for identifying protective factors and resilience mechanisms that help mitigate these disturbances. Therefore, this exploratory study aims to identify the most central (most highly connected) features of interoceptive sensibility and self-objectification in women with high and low levels of body dissatisfaction. Based on our previous research, we predict that relationships between interoceptive sensibility and self-objectification will be stronger in women with high body dissatisfaction compared to those with low body dissatisfaction, with differences in central features between the groups [34].

## Method

### Participants

Participants aged 18–75 were recruited online via Prolific.com as part of a larger cross-sectional study, which included a survey investigating body image and interoceptive sensibility, as well as a cognitive task (see Naraindas and Cooney, 2023 [34], for detailed methods and data collection procedures). This study includes a secondary analysis of the survey portion of the study. The survey received N = 1372 responses, with the sample comprising women aged 18–75 years who self-reported having no current eating disorder (See S1 Table for demographic information of all participants). For this secondary analysis, participants were categorized into high body dissatisfaction (N = 347) and low body dissatisfaction (N = 357) groups based on their scores on the Body Shape Questionnaire. Ethical approval for the study was granted

by the Human Research Ethics Committee (Humanities) at University College Dublin. All participants provided written informed consent, in line with the Declaration of Helsinki, and were informed of their right to withdraw from the study at any point. Participants were compensated at a rate of £8.50 per hour for their participation.

## Materials

The survey was created using Qualtrics [41]. Participants were provided with relevant information, consent forms, and debriefing materials within the survey.

## Measures

**Body image.** Body dissatisfaction and self-objectification were measured using validated self-report questionnaires. See Table 1. For details.

**Interoceptive sensibility.** Interoceptive sensibility was assessed using the Multidimensional Assessment of Interoceptive Awareness (MAIA-2) [33]. The scale comprises 37 items rated on a 5-point Likert scale, ranging from 'Never' to 'Always'. The scale assesses eight distinct dimensions via its respective sub-scales (See Table 2. for details).

## Data analysis

Descriptive statistics were calculated for all measures for the full sample (N = 1372). As relationships between interoceptive sensibility and body image total scores in the full sample have been analysed elsewhere (See: Naraindas and Cooney, 2023 [34]), this study will focus on the individual components of interoceptive sensibility and self-objectification in those in the high and low body dissatisfaction groups. Participants were divided into the lowest (25th percentile) and highest (75th percentile) quartiles based on their BSQ scores. Independent sample t-tests examined group differences in all questionnaire measures (see Table 3). Due to the significant difference between the groups in age (see Table 3), we examined age's influence on correlations between variables, similar to methods in other network analysis studies addressing demographic covariates as a sensitivity analysis [44] (see S2 Table. for raw dataset correlations and S3 Table. for correlations adjusted for age). The similarity between both tables led us to conduct the subsequent analyses on the raw dataset, without adjusting for age. Statistical analyses were conducted using R v4.3.1 [45].

**Table 1. Self-report questionnaires measuring body image, including a description of each questionnaire and the reliability scores for the current sample.**

| Questionnaire | Description | Reliability (Cronbach's α) |
|---|---|---|
| Body Shape Questionnaire (BSQ) [42] | The Body Shape Questionnaire (BSQ) measures concerns about body shape over the past four weeks using 34 items. For this study, a modified 30-item version was used, excluding four questions related to weight control and eating disorder behaviours. Higher BSQ scores indicate greater preoccupation with and dissatisfaction regarding one's body. In this study, the BSQ was used as a grouping variable. While the BSQ is validated for categorizing participants into high and low body dissatisfaction groups, the omission of four questions assessing eating disorder behaviours meant that the original validated cut-off scores could not be applied. To address this, we adopted an approach similar to other network analysis studies that utilize high/low group classifications [43]. Participants were divided into the lowest (25th percentile) and highest (75th percentile) quartiles based on their BSQ scores. Consistent with BSQ scoring guidelines for the 34-item version—ranging from no body dissatisfaction (scores below 80) to mild (80–110), moderate (110–140), and severe dissatisfaction (above 140) [42] — participants in this study appeared to align appropriately with these categories when grouped using the quartile split (see Table 3). | 0.97 |
| Objectified Body Consciousness questionnaire (OBC) [8]. | Two aspects of self-objectification, Body Shame and Body Surveillance, were measured using the subscales from the OBC. Each subscale contains 8 statements scored on a 7-point Likert scale from "strongly agree" to "strongly disagree". | 0.85 for body surveillance 0.81 for body shame |

 

**Table 2. MAIA-2 Subscales, cronbach's alpha and descriptions of subscales with an example question for each subscale.**

| Sub-scale | Description | Example Question | Reliability (Cronbach's α) |
|---|---|---|---|
| Noticing | The tendency to be conscious of one's body sensations, regardless of their comfort level. | When I am tense I notice where the tension is located in my body. | 0.71 |
| Not-Distracting | A behaviour where one does not ignore pain or uncomfortable body sensations. | I distract myself from sensations of discomfort. | 0.87 |
| Not-Worrying | Involves avoiding anxiety or distress about pain or discomfort. | I can remain calm and not worry when I feel discomfort or pain. | 0.47 |
| Attention Regulation | The ability to maintain focus on bodily sensations. | I can maintain awareness of my inner bodily sensations even when there is a lot going on around me. | 0.85 |
| Emotional Awareness | Recognition of the connection between emotional states and bodily sensations. | I notice how my body changes when I am angry. | 0.84 |
| Self-Regulation | The skill of managing distress through focused attention on bodily sensations such as breathing. | I can use my breath to reduce tension. | 0.79 |
| Listening (Body Listening) | The practice of actively listening to the body for insights. | I listen to my body to inform me about what to do. | 0.64 |
| Trusting | The extent to which one perceives their body as reliable. | I trust my body sensations. | 0.75 |

**Table 3. Mean scores (and SDs) of all scales, subscales, and comparison between High body dissatisfaction and Low body dissatisfaction groups.**

| Measure | Full Sample (N = 1372) Mean (SD) | High body dissatisfaction (N = 347) Mean (SD) | Low body dissatisfaction (N = 353) Mean (SD) | Independent samples t-test between low and high subgroups |
|---|---|---|---|---|
| **Age** | 40.71 (17.31) | 37.06 (16.02) | 43 (18.23) | $t$ = - 4.577, $p$ = < .001, d = - 0.346 |
| **BSQ Score** | 88.68 (33.87) | 136.04 (17.18) | 49.55 (7.88) | $t$ = - 85.82, $p$ = < .001, d = 6.488 |
| **MAIA-2 sub-scales** | | | | |
| Noticing | 2.27 (0.50) | 2.39 (0.45) | 2.17 (0.57) | $t$ = 5.46, $p$ = < .001, d = 0.413 |
| Not-distracting | 2.89 (0.75) | 2.74 (0.77) | 3.01 (0.78) | $t$ = -4.602, $p$ = < .001, d = - 0.348 |
| Not-worrying | 2.98 (0.65) | 2.77 (0.64) | 3.25 (0.61) | $t$ = -10.11, $p$ = < .001, d = - 0.764 |
| Attention Regulation | 3.06 (0.66) | 3.00 (0.65) | 3.20 (0.72) | $t$ = -3.905, $p$ = < .001, d = - 0.295 |
| Emotional awareness | 3.38 (0.86) | 3.48 (0.82) | 3.31 (0.87) | $t$ = 2.534, $p$ = 0.012, d = 0.192 |
| Self-Regulation | 2.96 (0.79) | 2.84 (0.79) | 3.16 (0.82) | $t$ = -5.192, $p$ = < .001, d = - 0.392 |
| Listening | 2.81 (0.81) | 2.42 (0.79) | 3.13 (0.78) | $t$ = -11.843, $p$ = < .001, d = - 0.895 |
| Trusting | 3.27 (0.87) | 2.78 (0.85) | 3.70 (0.83) | $t$ = -14.359, $p$ = < .001, d = - 1.085 |
| **Body Surveillance** | 33.30 (9.23) | 40.80 (7.19) | 26.42 (8.92) | $t$ = 23.454, $p$ = < .001, d = - 1.773 |
| **Body Shame** | 28.36 (9.25) | 37.99 (6.99) | 19.86 (6.10) | $t$ = 36.538, $p$ = < .001, d = 2.762 |

## Network analysis

To examine the network structures of the high and low body dissatisfaction groups, cross-sectional partial correlation networks were constructed using the R packages qgraph [46], bootnet [38], and glasso [47]. A network consists of 'nodes,' which are the key variables of interest (in this case, the subscales of the MAIA-2 and OBCS), and 'edges,' which are the connections between nodes (here represented by partial correlations). A Gaussian graphical model (GGM) was estimated for each network with extended Bayesian Information Criterion (EBIC) model selection, generating networks of nodes and edges. The networks were regularized using graphical LASSO (Least Absolute Shrinkage and Selection Operator)

to reduce spurious edges [47]. Network stability and edge weight accuracy were assessed using a case-dropping subset bootstrapping method (1000 iterations) with the bootnet package [38]. The correlation stability coefficient indicates the highest proportion of excluded cases while maintaining a correlation above 0.7 between the centrality indices of the full sample and subset samples, with a 95% confidence level [38]. Finally, we evaluated the significance of each node within the network using centrality metrics. We chose to present strength centrality, as it is usually more stable than other indices like closeness or betweenness [38]. Strength centrality refers to the sum of absolute edge weights directly connecting a node to others in the network; higher centrality scores indicate more connections to the node.

### Network comparison

To examine the differences between high and low body dissatisfaction subgroups, the network comparison test was performed using the R package Network Comparison Test (NCT) (version 2.2.1) [48]. We assessed global strength (the total sum of absolute connections between all node pairs), structure consistency (maximum difference between matrices of all edge strengths), and specific edge weight differences between the networks. Significance was determined using two-tailed permutation tests ($p < 0.05$) repeated 1000 times, comparing observed statistics with the reference distribution from the permutation procedure [48].

## Results

### Descriptive statistics

The mean scores and standard deviations of all scales/sub-scales for the full and grouped samples are reported in Table 3. Women in the high body dissatisfaction group had significantly higher scores on the BSQ, Body Surveillance, and body shame sub-scales compared to the low body dissatisfaction group. The high body dissatisfaction group also had significantly lower scores on all MAIA-2 sub-scales compared to the low body dissatisfaction group (see Table 3 for statistics).

### Network estimation

We estimated separate Gaussian Graphical Models (partial-correlation networks) for the high and low body dissatisfaction groups (see Fig 1.)

### Network stability and edge weight accuracy

The stability of strength centrality estimates exceeded cut-offs to be considered interpretable for both high (CS(cor = .75) =.67) and low (CS(cor = .70) =.67) body dissatisfaction samples. For edge weight accuracy, confidence interval widths were moderate and overlapped for some edges, indicating that while not all edge weights differ, a substantial number likely do (see S1 Fig for edge weight accuracy plots for high and low body dissatisfaction groups).

### Centrality indices

Centrality indices for the high and low body dissatisfaction networks are shown in Fig 2. The node highest in centrality in the high body dissatisfaction network was Listening, while in the low body dissatisfaction network it was Emotional Awareness. The least central node in both networks was Not Distracting. Fig 3. shows the bootstrapped difference tests and significant differences in strength centrality between nodes within the high and low body dissatisfaction networks. In the high body dissatisfaction network, Listening was significantly more central than the Not Distracting, Noticing, Not Worrying nodes of MAIA-2 and the Body Shame and Body Surveillance nodes of the objectified body consciousness scale. However, it did not show significantly greater centrality than Self-Regulation, Trusting, Emotional Awareness, and Attention Regulation nodes. In the low body dissatisfaction network, Emotional Awareness was significantly more central than Not

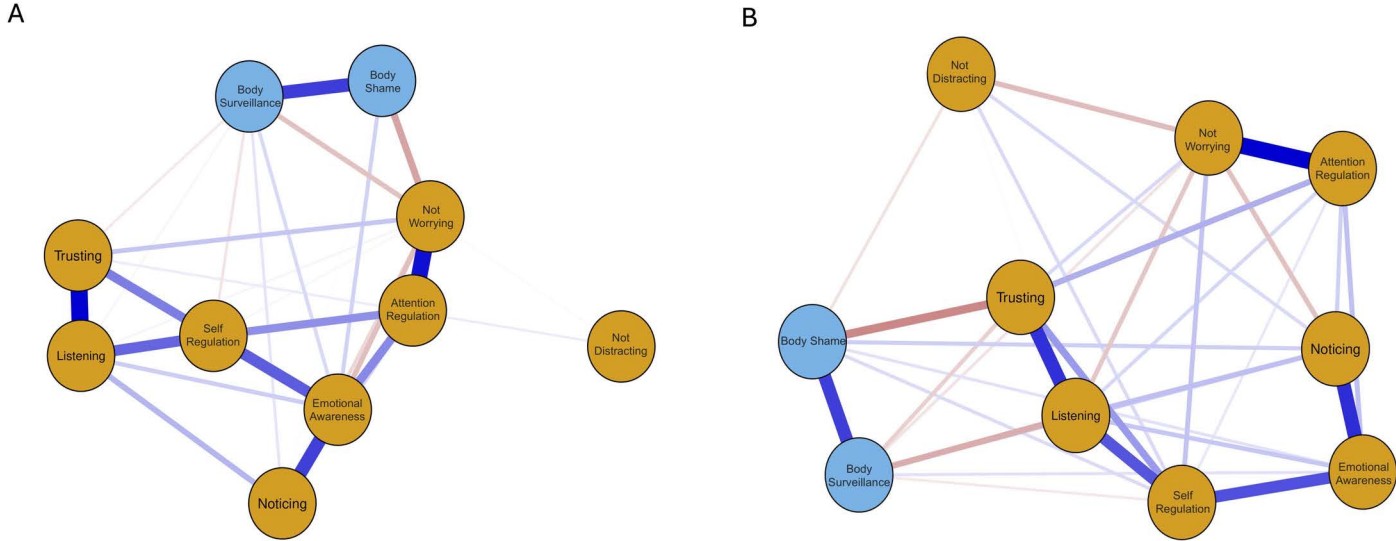

● Interoceptive Sensibility (MAIA–2)
● Self Objectification (OBCS)

**Fig 1. Estimated Regularized Partial Correlation Network of body shame, Body Surveillance and interoceptive sensibility scores among women with High (A) and Low (B) body dissatisfaction.** The calculated total subscale scores of each MAIA-2 sub-scale and the OBC sub-scale for all cases are used as the "nodes" in the network. The "edges'' connecting these nodes represent the potential associations, with partial correlations ranging from -1 to 1. Positive edges are depicted by blue lines, while negative edges are shown in red. The thickness of an edge indicates the strength of the connection, as measured by the partial correlation.

Distracting, Noticing, and Trusting nodes of MAIA-2, and the Body Shame and Body Surveillance nodes of objectified body consciousness scale, but not significantly higher than Not Worrying, Attention Regulation, Self-Regulation, and Listening nodes.

## Network comparison test

We observed network differences between the high and low body dissatisfaction groups. The high body dissatisfaction network was more connected (75%; 34/45 non-zero edges) than the low body dissatisfaction network (64%; 29/45 non-zero edges), indicating a denser network. The high body dissatisfaction network also had more negative relationships between interoceptive sensibility and self-objectification nodes than the low body dissatisfaction network. The NCT results showed a significant difference between the network structures (M=0.21, p=0.026), but global strength was comparable (S = 0.65, p=0.10). Seven edges differed significantly between the networks. In the high body dissatisfaction group, there were stronger negative relationships between Body-Surveillance and Listening (p=0.014), Trusting and Body Shame (p<0.01), Not-distracting and Not-worrying (p=0.01), and Not-worrying and Listening (p<0.001). There were stronger positive relationships between Attention Regulation and Trusting (p=0.024). In the low body dissatisfaction group, there were stronger negative relationships between Not-worrying and Body Shame (p=0.01) and stronger positive relationships between Attention Regulation and Self-Regulation (p=0.03).

## Discussion

This study reveals significantly different relationships between features of interoceptive sensibility and self-objectification among women with high body dissatisfaction compared to women with low body dissatisfaction. Interestingly, the network

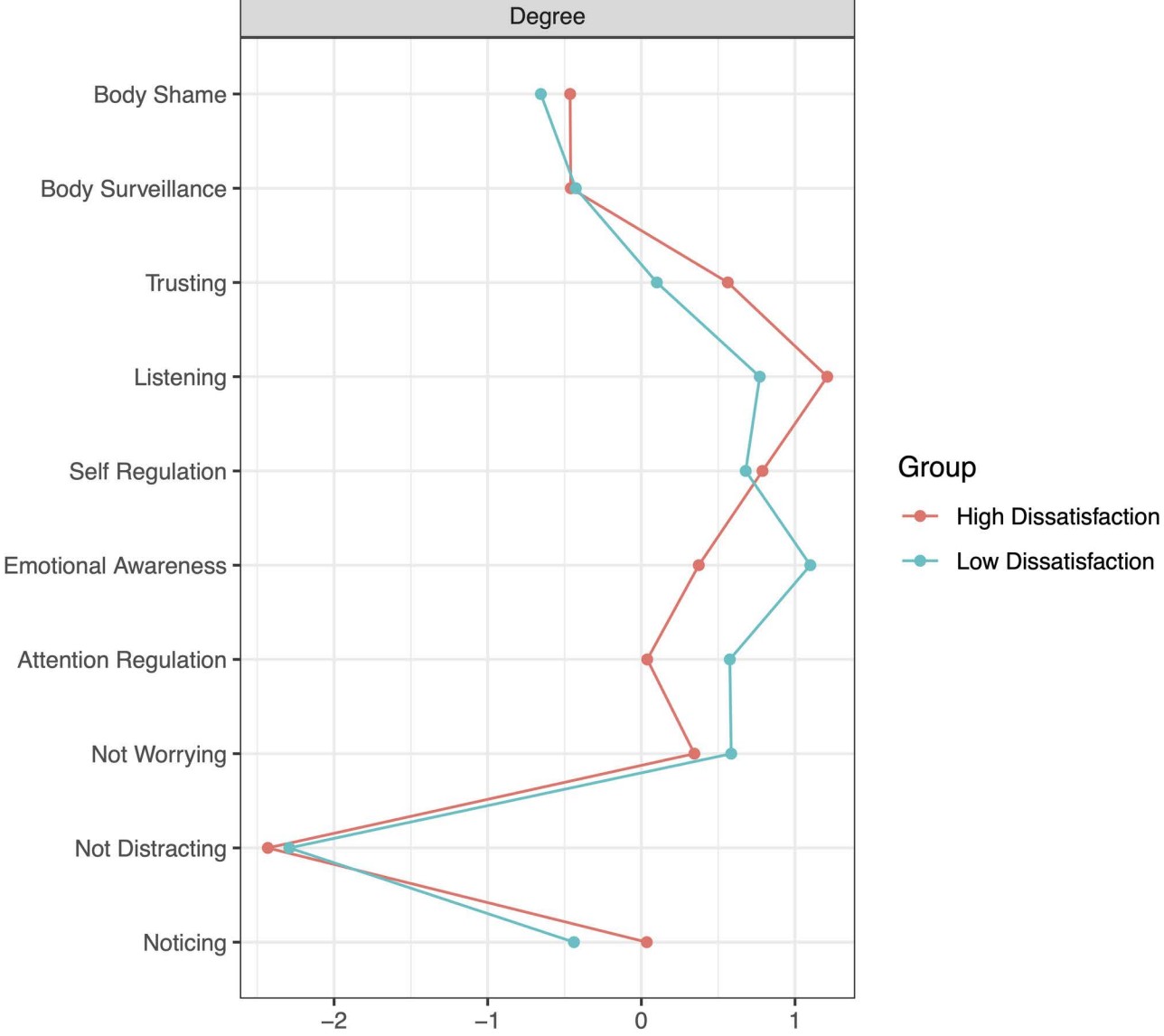

**Fig 2. Network strength degree for self-objectification and interoceptive sensibility nodes in the high and low body dissatisfaction groups.** Higher values indicate greater connections to the node.

analysis demonstrated that it was not self-objectification, but features of interoceptive sensibility that were most central to high and low body dissatisfaction.

Specifically, we found that relationships between interoceptive sensibility and self-objectification were stronger and more negatively related in the high body dissatisfaction group (i.e., a denser network) compared to the low body dissatisfaction group. This supports the research that indicates that women with higher levels of self-objectification tend to be less attuned to their bodily states [18,49,50]. According to network theory, the spreading of activation from one node to another is easier in denser networks [51]. Indeed, the high body dissatisfaction group exhibited significantly stronger negative relationships between Body surveillance and Listening, as well as between Body Shame and Trusting, compared to the low body dissatisfaction group. Additionally, low levels of Listening, Self-Regulation, and Trusting were central features

## A. High Dissatisfaction

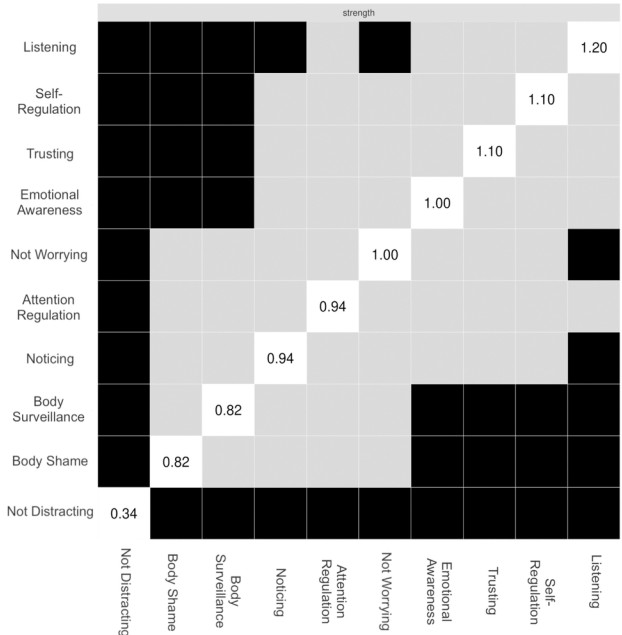

## B. Low Dissatisfaction

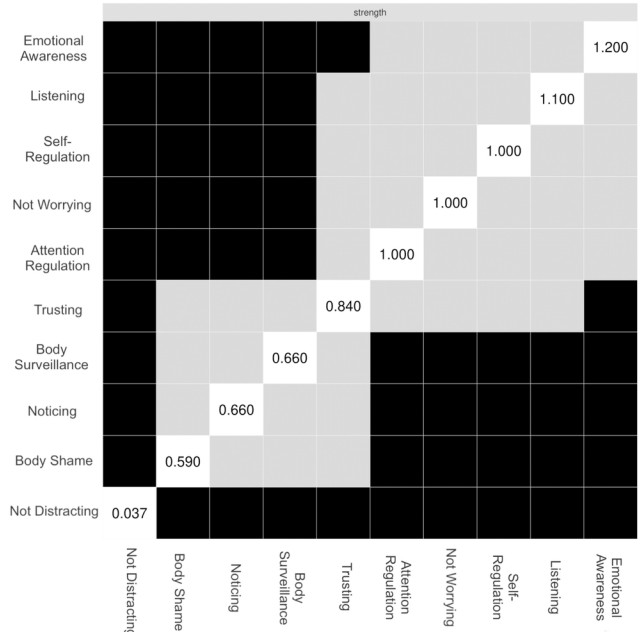

**Fig 3. Bootstrapped differences in strength centrality.** Plot A shows significant differences between the strength centrality of the nodes in the high body dissatisfaction group. Plot B shows the same information for the low body dissatisfaction group. Grey squares represent nodes that did not show significant differences from each other, while black squares signify nodes that did differ significantly. White squares display the strength centrality values for each node.

of the high body dissatisfaction network. This means that these features are highly interconnected, creating a dynamic of interdependence where changes in one node are more likely to affect others connected to it [52]. Taken together, these findings align with the competition of cues hypothesis [20], indicating that due to limited attentional capacity, women who engage in self-objectification and focus strongly on their external body have fewer attentional resources to process interoceptive signals.

The finding of low Trusting being central to women with high body dissatisfaction in particular is an interesting finding. Research indicates that low body trust is strongly associated with poor mental health outcomes and shows a robust negative correlation with suicidal ideation and attempts [53,54]. It not only predicts the severity and longitudinal progression of suicidal ideation and self-harm [55] but also moderates the relationship between exercise dependence and suicidal thoughts [56]. Additionally, women with high body dissatisfaction often engage in increased dieting and restrictive eating, suggesting they may be less attuned to their internal bodily signals [57]. Furthermore, low body trust has been consistently associated with heightened concerns about shape and weight, making it the most reliable interoceptive sensibility predictor of body dissatisfaction [36,58,59]. Other network analysis studies have shown that experiencing body sensations as unsafe or untrustworthy significantly bridges eating disorder symptoms related to weight and shape dissatisfaction, suggesting that low body trust is a common feature of both body dissatisfaction and eating pathology [58].

Additionally, the stronger association between low Trusting and Body Shame in the high body dissatisfaction network may result from these individuals distancing themselves from their bodies due to the shame experienced from it. This distancing can lead to perceiving internal bodily signals as unsafe [60]. For instance, women with high body dissatisfaction may ignore hunger signals or misinterpret them as untrustworthy due to perceived associations of these signals with

weight gain [58]. Therefore, enhancing body trust may be crucial for improving the health and well-being of women with high body dissatisfaction.

Alternatively, high levels of Emotional Awareness, Listening, and Self-Regulation were central features of the low body dissatisfaction network. There were also stronger negative relationships between the Not-Worrying and Body Shame subscales in the low group compared to the high group, indicating that women with lower body shame were less likely to worry and catastrophise about bodily discomfort. Indeed, research suggests that women with healthier body image are more attuned to their physical needs [61] and more likely to participate in self-care and mindfulness behaviours [62]. This could underly the high levels of body Listening and Self-Regulation that were central to the low body dissatisfaction group. Also, considering that the most central node in the low body dissatisfaction group was high emotional awareness, this suggests that better management of emotional states toward the body might be key to lower levels of body dissatisfaction. This aligns with findings from Todd et al. (2019), who found that the MAIA-2 Emotional Awareness subscale positively predicted body appreciation, indicating that improved emotional awareness regarding bodily sensations may contribute to a positive body image [36].

There were also differences in the way features of interoceptive sensibility connected to each other between the high and low body dissatisfaction groups. In the high body dissatisfaction group, the relationships in interoceptive sensibility were less consistent than in the low body dissatisfaction group. For example, whilst some features were negatively related (e.g., Not-Distracting and Not-worrying, Not-worrying and Listening), others were positively related (e.g., Attention Regulation and Trusting). In contrast, the low body dissatisfaction group and general population samples from other studies predominantly show positive relationships among the MAIA-2 subscales [33,63]. However, it is important to highlight that the Not-Distracting and Not-Worrying sub-scales of the MAIA-2 have demonstrated low reliability and internal consistency across various studies [33,36,64]. In this study, the Not-Worrying sub-scale showed the lowest reliability among all MAIA-2 scales, while the Not-Distracting sub-scale appeared to be the least central to the high and low body dissatisfaction networks. Consequently, future research may consider excluding these sub-scales, particularly in body image studies. Despite this, the current study provides preliminary evidence of altered connectivity patterns in interoceptive sensibility among women with high body dissatisfaction. Both altered interoception and body image are often considered nonspecific risk factors for negative physical and mental health outcomes in women [13,65,66]. Indeed, women report poorer body image during periods of turbulent and altering inner body experiences (e.g., menopause [67], menstruation [68] and pregnancy [15,69]. Therefore, future research should investigate how altered interoception may contribute to poor body image during changes in women's health across the lifespan.

## Conclusions and limitations

Overall, the network analysis revealed distinct connectivity patterns between features of interoceptive sensibility and self-objectification in women with high and low body dissatisfaction. However, a limitation of this study is that we only focused on one aspect of interoception, namely self-report interoceptive sensibility. Interoception is a multidimensional construct comprising interoceptive accuracy, interoceptive sensibility, and interoceptive awareness, according to the tripartite model of interoception [70]. However, these different constructs are not proxies for one another, and numerous studies have demonstrated independence between subjective self-report measures of interoception (i.e., interoceptive sensibility) and objective measurements of interoception (e.g., interoceptive accuracy) [71–73]. Therefore, future research should investigate relationships between self-objectification and other interoception dimensions to better understand these relationships within the context of body dissatisfaction.. Furthermore, the present findings were based on cross-sectional data; thus, we cannot infer that the differences noted above between the high and low body dissatisfaction groups are due to body dissatisfaction alone.

For example, given the diverse nature of the sample, cultural differences in body dissatisfaction and interoceptive sensibility may have influenced the findings. For instance, individuals from non-Western cultures demonstrate higher

interoceptive sensibility [74] and may express body dissatisfaction in culturally specific ways, such as concerns with facial features [75] or skin tone [76]. Nonetheless, cross-cultural research suggests that the antagonistic relationship between body image and interoception is observed across diverse ethnic groups [37,77]. Additionally, the wide age range of the sample may have introduced age-related effects, as younger individuals often report greater appearance-based body dissatisfaction due to societal pressures [78], while older individuals may experience more dissatisfaction with their body functionality as opposed to appearance [79].

Given the complex nature of body image and interoceptive sensibility, future longitudinal studies should consider using network analysis to examine their dynamic relationships throughout the lifespan. This is particularly important during adolescence and emerging adulthood, which are critical periods for the development of eating disorders [80].

Using network analysis, this study has identified important features of interoceptive sensibility, such as diminished body listening, trust, and emotional regulation, that are central to body dissatisfaction. Future body image interventions could benefit from focusing on enhancing individuals' ability to attune to and trust their bodily sensations, alongside improving emotional regulation. Such approaches may lead to more effective outcomes in promoting health and well-being among women with high body dissatisfaction.

## Supporting information

**S1 Fig. Edge weight accuracy plot for high and low body dissatisfaction group.**
(DOCX)

**S1 Table. Demographic information for all participants (N = 1372) in the study.**
(DOCX)

**S2 Table. Spearman-product correlation matrix for raw dataset.**
(DOCX)

**S3 Table. Spearman-product correlation matrix on age regressed dataset. To control for the effect of age, each variable was regressed by age and the obtained standardized residuals were correlated.**
(DOCX)

## Author contributions

**Conceptualization:** Akansha Mahesh Naraindas, Amy McInerney, Sonya Deschênes, Sarah Maeve Cooney.

**Data curation:** Akansha Mahesh Naraindas.

**Formal analysis:** Akansha Mahesh Naraindas.

**Funding acquisition:** Sarah Maeve Cooney.

**Investigation:** Akansha Mahesh Naraindas, Amy McInerney, Sonya Deschênes, Sarah Maeve Cooney.

**Methodology:** Akansha Mahesh Naraindas, Sarah Maeve Cooney.

**Project administration:** Akansha Mahesh Naraindas, Sarah Maeve Cooney.

**Resources:** Akansha Mahesh Naraindas, Sarah Maeve Cooney.

**Software:** Akansha Mahesh Naraindas.

**Supervision:** Sarah Maeve Cooney.

**Visualization:** Akansha Mahesh Naraindas.

**Writing – original draft:** Akansha Mahesh Naraindas.

**Writing – review & editing:** Akansha Mahesh Naraindas, Amy McInerney, Sonya Deschênes, Sarah Maeve Cooney.

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
