## [Decision Letter · Decision Letter 0]

25 Mar 2025

PONE-D-24-55081Differences in the Relationships Between Interoceptive Sensibility and Self-Objectification in Women with High and Low Body Dissatisfaction: A Network AnalysisPLOS ONE

Dear Dr. Mahesh Naraindas,

Thank you for submitting your manuscript to PLOS ONE. After careful consideration, we feel that it has merit but does not fully meet PLOS ONE’s publication criteria as it currently stands. Therefore, we invite you to submit a revised version of the manuscript that addresses the points raised during the review process.

We look forward to receiving your revised manuscript.

Kind regards,

Silvana Mula, Ph.D.

Academic Editor

PLOS ONE

Journal Requirements:

“This project was funded by the University College Dublin Ad Astra research fellowship (ID: C776AAB4) to S.M.C. and associated Ad Astra PhD scholarship to A.M.N.”

4. Please note that your Data Availability Statement is currently missing the repository name and/or the DOI/accession number of each dataset OR a direct link to access each database. If your manuscript is accepted for publication, you will be asked to provide these details on a very short timeline. We therefore suggest that you provide this information now, though we will not hold up the peer review process if you are unable.

Reviewers' comments:

Reviewer's Responses to Questions

**Comments to the Author**

1. Is the manuscript technically sound, and do the data support the conclusions?

Reviewer #1: Yes

Reviewer #2: Yes

2. Has the statistical analysis been performed appropriately and rigorously? 

Reviewer #1: Yes

Reviewer #2: Yes

3. Have the authors made all data underlying the findings in their manuscript fully available?

Reviewer #1: Yes

Reviewer #2: Yes

4. Is the manuscript presented in an intelligible fashion and written in standard English?

Reviewer #1: Yes

Reviewer #2: Yes

5. Review Comments to the Author

Reviewer #1: It is an interesting article which reaches interesting conclusions.

The limitations of the study that I observed (for example, the possible differences in the results considering age groups or the different aspects of interoception) are pointed out in the limitations of this study, therefore, they are rectified. However, it would have been interesting to point out also in the limitations the possible cultural differences that could influence interoception and body dissatisfaction, since it is known that there is a cultural influence on this and the sample is composed of women from different countries. Also to report in a table the demographic data of the participants considering the above mentioned.

These findings open the door to new studies on the subject and the clinical management of body dissatisfaction.

Reviewer #2: Thank you for the opportunity to review the article entitled "Differences in the Relationships Between Interoceptive Sensibility and Self-Objectification in Women with High and Low Body Dissatisfaction: A Network Analysis". The work is well written and organized, focusing on a central theme currently. I believe the work is of high quality. I will only highlight two aspects regarding limitations and future directions of research. The sampling through a rewarded platform, which uses only self-report questionnaires (especially for self-report of absence of eating disorders and for interoceptive abilities) is a limitation. Interoception is a complex construct and, although a widely used and validated instrument was used, it may not be best captured in a self-report. Furthermore, the age range of the sample is very wide. Young adults may have different social pressures than older adults, and there may be different nuances. It would be interesting to investigate these aspects in the future also in a younger sample: adolescents and emerging adulthood. Overall I think the work is very interesting and, after adding these limitations, it can be an important contribution to the literature.

6. PLOS authors have the option to publish the peer review history of their article (what does this mean? ). If published, this will include your full peer review and any attached files.

**Do you want your identity to be public for this peer review?** For information about this choice, including consent withdrawal, please see our Privacy Policy .

Reviewer #1: No

Reviewer #2: No

---

## [Author Response · Author response to Decision Letter 0]

3 Apr 2025

Reviewer comments have been uploaded as a word document as an attached file.

Editor comments responses:

1. I have put the document in PLOS one style as per the website provided

2. Funding statement has been removed from the manuscript

3. Regarding the financial disclosure statement 'The funders had no role in study design, data collection and analysis, decision to publish, or preparation of the manuscript.' is appropriate

4. The data is publicly available on OSF via this link here: https://osf.io/25cjh

5. The full ethics statement is now available in the 'methods' section of the manuscript as stated here:Ethical approval for the study was granted by the Human Research Ethics Committee (Humanities) at University College Dublin. All participants provided written informed consent, in line with the Declaration of Helsinki, and were informed of their right to withdraw from the study at any point.

6. The reference list is complete and correct; no retracted papers have been cited

---

## [Editor Report · Decision Letter 1]

10 Apr 2025

Differences in the Relationships Between Interoceptive Sensibility and Self-Objectification in Women with High and Low Body Dissatisfaction: A Network Analysis

PONE-D-24-55081R1

Dear Dr. Naraindas,

We’re pleased to inform you that your manuscript has been judged scientifically suitable for publication and will be formally accepted for publication once it meets all outstanding technical requirements.

Kind regards,

Silvana Mula, Ph.D.

Academic Editor

PLOS ONE
---

## [Editor Report · Acceptance letter]

PONE-D-24-55081R1

PLOS ONE

Dear Dr. Mahesh Naraindas,

I'm pleased to inform you that your manuscript has been deemed suitable for publication in PLOS ONE. Congratulations! Your manuscript is now being handed over to our production team.

Kind regards,

on behalf of

Dr. Silvana Mula

Academic Editor

PLOS ONE